# Rootstocks Influence Yield Precocity, Productivity, and Pre-Harvest Fruit Drop of Mandared Pigmented Mandarin

**Marco Caruso** [1,*] **, Alberto Continella** [2,*] **, Giulia Modica** [2] **, Claudia Pannitteri** [2] **, Riccardo Russo** [1,2] **, Fabrizio Salonia** [1,2] **, Carmen Arlotta** [1] **, Alessandra Gentile** [2] **and Giuseppe Russo** [1]

[1]   CREA, Research Centre for Olive, Fruit and Citrus Crops, Corso Savoia 190, 95024 Acireale, Italy; riccardo.russo.1991@gmail.com (R.R.); fabrizio.salonia@unict.it (F.S.); carmen.arlotta@crea.gov.it (C.A.); giuseppe.russo@crea.gov.it (G.R.)

[2]   Department of Agriculture, Food and Environment (Di3A), University of Catania, Via Valdisavoia 5, 95123 Catania, Italy; giulia.modica@unict.it (G.M.); cla.pannitteri@hotmail.it (C.P.); gentilea@unict.it (A.G.)

*   Correspondence: marco.caruso@crea.gov.it (M.C.); acontine@unict.it (A.C.)

**Abstract:** Citrus fruit quality and scion productivity are influenced by the choice of rootstock. We aimed to evaluate the effect of rootstocks on yield and fruit quality of Mandared, a triploid pigmented mandarin. To do so, we established a rootstock field trial on a high pH soil (8.6) in which Mandared was grafted onto 11 rootstocks. These included some standard rootstocks, such as trifoliate orange ((*Poncirus trifoliata* (L.) Raf.), Troyer citrange (*Citrus sinensis* (L.) Osb. × *P. trifoliata*), Swingle citrumelo (*Citrus paradisi* Macf. × *P. trifoliata*), and C35 citrange (*C. sinensis* × *P. trifoliata*), as well as new releases from the Council for Agricultural Research and Economics (CREA, Acireale, Italy) and the University of California Riverside (UCR). The cumulative yield was measured over five consecutive years, while fruit quality was analyzed for two years. The trees on C35, C57 (*Citrus sunki* Hort. ex. Tan. × *P. trifoliata*), and C22 (*C. sunki* × *P. trifoliata*), started to set fruits one year earlier than the others. The trees on C57 provided some of the highest cumulative yields and canopy volumes. The production of Mandared grafted onto C57 was double that of Mandared grafted onto Troyer, while Mandared grafted onto C35 and C22 resulted in the best yield efficiency. The trees on Swingle and C57 significantly reduced the pre-harvest fruit drop, to which Mandared is particularly sensitive. However, grafting Mandared onto Swingle resulted in the highest variation among replicates, probably due to its high sensitivity to iron chlorosis. Most of the fruit quality parameters, such as fruit size, total soluble solids (TSS), and acidity were not significantly different among the rootstock treatments. However, fruits produced by Mandared grafted onto C22 had one of the highest rates of anthocyanin accumulation. The results indicate that C57, C35, and C22 were the most suitable rootstocks for Mandared in South-Eastern Sicily.

**Keywords:** anthocyanins; citrus; fruit quality; rootstock/scion combination; yield efficiency

## 1. Introduction

Rootstocks are known to affect the performance of many traits of different citrus varieties, including tolerance to biotic and abiotic stresses, fruit quality and size, productivity, ripening period, and yield precocity [1,2]. Recently, an effect of rootstocks on anthocyanin pigmentation of blood oranges was also observed [3]. Therefore, in the case of pigmented citrus cultivars, an appropriate rootstock choice is essential to produce high quality fruits.

Italy is the second largest European producer of citrus fruits [4]. Most of the citrus fruits produced in Italy are intended for fresh sale at markets, and therefore fruit quality is of great importance for the Italian citrus fruit production industry. Until the early 2000s, most of the Italian citriculture was almost exclusively based on the use of sour orange as a rootstock, which conferred high quality to the scion cultivars. After the outbreak of citrus tristeza virus (CTV) [5], new plantings have been established, mostly using Troyer and Carrizo citranges (*Citrus sinensis* (L.) Osb. × *Poncirus trifoliata*), and Swingle citrumelo (*Citrus paradisi* Macf. × *P. trifoliata*). However, these rootstocks are known to have low tolerance to high pH calcareous soils, which are typical of many growing areas of Southern Italy. As such, growers prefer to use Volkamer lemon (*Citrus volkameriana* Pasq.) or Alemow (*Citrus macrophylla* Wester), which guarantee high yield but an overall low fruit quality and low anthocyanin pigmentation [6]. New rootstocks are therefore needed to confer high productivity and high fruit quality to citrus fruits grown under suboptimal pedological conditions.

Recently, many breeding programs around the world have released new rootstocks [2,7]. Among these, the University of California Riverside (UCR) released three Sunki mandarin (*Citrus sunki* Hort. ex Tan.) × trifoliate orange (*P. trifoliata*) hybrid rootstocks, namely Bitters (C22), Carpenter (C54), and Furr (C57). These three rootstocks showed good performance in many trials in terms of yield efficiency (C22), productivity (C54 and C57), and high (C22) or moderate (C54 and C57) tolerance to calcareous soils [8]. The Council for Agricultural Research and Economics (CREA, Italy) released three *Citrus latipes* (Swing.) Tan. × *P. trifoliata* hybrid rootstocks (F5P12, F6P12, and F6P13) that showed high cumulative yield in combination with TDV Tarocco blood orange, Washington Navel orange, and SRA92 Clementine (*Citrus clementina* Hort. ex Tan.) [9].

Mandared (Figure 1) is a triploid hybrid between Nules Clementine and tetraploid Tarocco orange [10].

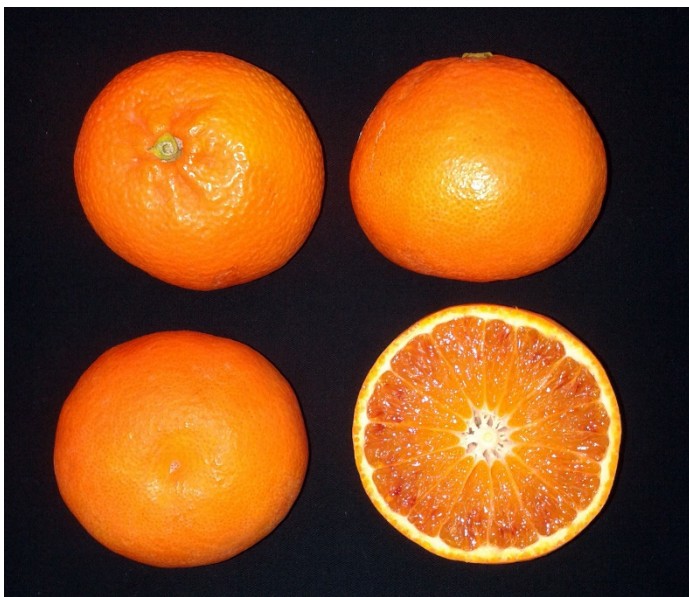

**Figure 1.** Fruits of Mandared (*Citrus clementina* 2 x × *Citrus sinensis* 4 x).

Mandared was released by CREA in 2004 and is cultivated or under evaluation in different citrus producing countries. It ripens from mid-January to mid-February in the northern hemisphere. Mandared has unique pomological features among seedless mandarins, including its pulp pigmentation, which has attracted the interest of many growers. It is a mid to large size mandarin, with an oblate shape and a thin and smooth rind, and an acidic taste resembling that of blood oranges [6]. In suitable environments, the flavor of Mandared fruits reaches a good balance between sugars and acids. However, after almost a decade since the establishment of the first Mandared commercial plantings,

some disadvantages have become evident. These include: a delay in the first production compared to other citrus varieties; low productivity; thorniness, which causes damage to fruit rinds, especially in windy areas; the smooth and thin rind, which does not protect fruits from damage; and a high incidence of pre-harvest fruit drop, which is probably the largest problem (G. Russo, personal communication). Mandared fruits tend to fall before they reach commercial maturity, with a high percentage of fruit loss compared to other citrus cultivars, especially in suboptimal climatic conditions (frost, dry winds, etc.).

Here we describe the performance of Mandared triploid mandarin grafted onto 11 rootstocks and evaluated during five consecutive years. The trial was performed: (i) to assess the influence of the rootstock on Mandared fruit quality (including pigmentation) and productivity; and (ii) to search for the most suitable CTV-tolerant rootstocks that can be used as alternatives to sour orange for the establishment of Mandared orchards under the soil and climate conditions of South-Eastern Sicily.

## 2. Materials and Methods

### 2.1. Plant Material and Trial Management

The trial was conducted at the CREA experimental farm of Palazzelli (Siracusa, Italy) (37° 20′ N, 14° 53′ E, 48 m a.s.l.), with trees spaced at 6 × 4 m. Mandared (*C. clementina* × *C. sinensis*) budsticks were grafted onto 11 two-year old rootstocks (for details, see Table 1) in the spring of 2009.

**Table 1.** List of rootstocks used in the field trial.

| Rootstock Name | Botanical Species | Abbreviation |
|---|---|---|
| Bitters (C22) citrandarin | *Citrus sunki* Hort. ex. Tan. × *Poncirus trifoliata* | C22 |
| C35 citrange | *Citrus sinensis* (L.) *Osb.* × *P. trifoliata* (L.) *Raf.* | C35 |
| Carpenter (C54) citrandarin | *C. sunki* × *P. trifoliata* | C54 |
| Furr (C57) citrandarin | *C. sunki* × *P. trifoliata* | C57 |
| Swingle citrumelo | *Citrus paradisi* Macf. × *P. trifoliata* | Swingle |
| 68IG26-C1F6-P12 | *Citrus latipes* (Swing.) Tan. × *P. trifoliata* | F6P12 |
| 68IG26-C1F6-P13 | *C. latipes* × *P. trifoliata* | F6P13 |
| Serra trifoliate orange | *P. trifoliata* | *P. trifoliata* |
| *Severinia buxifolia* | *Severinia buxifolia* (Poir.) Ten. | *S. buxifolia* |
| Troyer citrange | *C. sinensis* × *P. trifoliata* | Troyer |
| Flying dragon | *P. trifoliata* | F. dragon |

The trial was established in July 2010 in a completely randomized design using 10 single tree replicates for each rootstock. Soil conditions were as described in Caruso et al. [11], with pH 8.6 and active lime ranging from 23.5 to 38.9 g kg$^{-1}$ in different parts of the experimental plot.

Plants were grown under conventional cultural practices. Fruits were treated with Spinosad (Spintor Fly) every 10 days from September to mid-November to control Mediterranean fruit fly (*Ceratitis capitata*). Moreover, fruits were treated once with 20 ppm 3,5,6-trichloro-2-pyridyl-oxyacetic acid (Tryclopir) at color break (mid-November), and a month later with 15 ppm 2, 4-dichlorophenoxy acetic acid (2, 4-D), to prevent or reduce fruit drop.

### 2.2. Sampling

Fruits were sampled every second half of January from 2015 to 2019. To measure the yield, the production of each replicate tree was harvested and weighed, and all fruits were counted. Average fruit weight was calculated by dividing the total sample weight by the number of fruits per tree. Due to the sensitivity of Mandared to pre-harvest fruit drop, two rounds of fruit drop counts were performed every year before harvest (10 days and one day before, respectively). The fruit drop rate was calculated as the ratio between the number of fruits that dropped before harvest and the total number of fruits, and multiplying by 100. Cumulative fruit drop did not include the 2015 harvest since not all rootstocks produced fruits in 2015. Dropped fruits were included in the yield. Canopy volume

was approximated as one half prolate spheroid, as suggested by Turrel [12], with V = $4/6\pi h(d/2)^2$, where h is tree height, and d is tree diameter. Yield efficiency of the canopy was calculated as the ratio of the cumulative yield to the canopy volume in the spring of 2018.

### 2.3. Fruit Quality

Samples of 15 fruits per plant were collected in 2017 and 2019 from three representative trees per rootstock to evaluate the rootstock effect on fruit quality traits. Specifically, we evaluated the following parameters: juice yield, peel color index (CI); total soluble solids (TSS); titratable acidity (TA); and total anthocyanin concentration.

Juice was extracted from the fruits using a domestic squeezer (Citrus Juicer JE290, Kenwood, UK) and filtered before analysis. Three juice samples, from the pooled juice of five fruits from three replicates per rootstock combination, were used for chemical analyses. Juice yield was calculated as follows: (Juice weight/fruit weight) $\times$ 100. TSS were measured using a digital refractometer (Atago CO., LTD, model PR-32 $\alpha$, Tokyo, Japan), with the results expressed as °Brix. TA, expressed as the percentage of anhydrous citric acid, was determined using a potentiometric titration (Hach, TitraLab AT1000 Series) of the juice with 0.1 N NaOH beyond pH 8.1, according to the AOAC method [13] with the results expressed as g $L^{-1}$ of citric acid equivalent. Ripening index (RI) was calculated as the ratio between TSS and TA.

Peel color was recorded on two opposite points of the equatorial region for each fruit using a Minolta CR-400 chroma-meter (Minolta Corp., Osaka, Japan). The CI, which is widely used in the citrus industry as a maturation index, was calculated following the method described by Jiménez-Cuesta et al. [14], and was determined as 1000 a*/L*b*, where L* is lightness, a* is the red–green component, and b* is the yellow–blue component. Juice total anthocyanin concentration was measured using a Nanodrop (NanoDrop 2000, Thermo Scientific) spectrophotometer (at 510 and 700 nm) using the pH differential method [15], and was expressed as cyanidin 3-glucoside equivalents (mg $L^{-1}$).

### 2.4. Data Analysis

Comparisons of means were performed using one-way analysis of variance (ANOVA) with Tukey tests and the confidence level was at 95%, with rootstock genotypes as fixed effects. Statistical analysis was performed using the R Commander [16] package or R software, version 3.6.1. A radar chart was generated using Poltly chart studio (https://chart-studio.plot.ly) to summarize the productive performance of the analyzed rootstocks. In particular, cumulative yield, yield efficiency, and cumulative fruit drop were independently rescaled within the range (0, 1) using maximum and minimum normalization on the basis of the best (1) and the worst (0) performance for each parameter.

## 3. Results

In the period of time between the establishment and the end of the field trial, some replicates died or showed poor growth. Specifically, six Mandared plants grafted onto F. dragon, five plants grafted onto *S. buxifolia*, and one plant grafted onto F6P13 died during the first 3–4 years after planting. The remaining Mandared plants grafted onto F. dragon did poorly in terms of production during all years of the trial, and for this reason were excluded from further analysis.

Moreover, some non-representative plants were not considered in the statistical analysis. We excluded one C22, one F6P12, and two Swingle replicates, since the productivity of Mandared grafted onto these rootstocks, and their canopy volume were extremely low compared to the other replicates. This might be due to the presence of off-types that were not recognized during plant propagation. The statistical analysis regarding cumulative production, yield efficiency, percent fruit drop, and quality traits were performed using the rest of the replicates.

Iron chlorosis symptoms were observed on trees on F. Dragon, *P. trifoliata*, and Swingle, but not in the rest of the plants. Most of the plants started to set fruits in 2015 (five years after planting), with significant differences among trees grafted onto different rootstocks. A shorter non-productive

period was observed in Mandared grafted onto C35, C57, and C22, with means of 11.7, 9.6, and 8.8, respectively. Mandared onto C54 produced 5.2 kg of fruits per tree, while the trees on the other rootstocks had very low (around 1 kg of fruit produced per plant or less) or no fruit production in 2015 (Table 2).

**Table 2.** Yield of Mandared grafted onto 10 rootstocks calculated from 2015 to 2019. Different letters indicate significantly different means according to Tukey's test at $p < 0.05$.

| Rootstock | Yield (kg Per Tree) | | | | |
|---|---|---|---|---|---|
| | 2015 | 2016 | 2017 | 2018 | 2019 |
| C22 | 8.8 ab | 48.9 ac | 39.1 ab | 40.7 ab | 45.3 ab |
| C35 | 11.7 a | 71.9 a | 34.2 ac | 43.1 a | 40.6 ab |
| C54 | 5.2 ac | 49.3 ac | 46.6 ab | 43.1 a | 51.0 ab |
| C57 | 9.6 a | 68.2 ab | 46.8 ab | 49.0 a | 51.5 ab |
| F6 P12 | 1.1 c | 45.6 bcd | 25.5 bc | 37.6 abc | 40.9 ab |
| F6P13 | 0.8 c | 20.5 def | 35.9 ac | 29.6 ad | 56.6 a |
| *P.trifoliata* | 0.4 c | 14.0 ef | 43.2 ab | 8.5 d | 39.0 ab |
| *S. buxifolia* | 0.1 c | 2.8 f | 9.0 c | 12.0 cd | 24.9 ab |
| Swingle | 1.2 bc | 38.3 ce | 51.6 a | 31.3 ad | 59.8 a |
| Troyer | 0.9 c | 12.3 ef | 54.5 a | 18.5 bd | 25.3 b |

Cumulative production between 2015 and 2019 (Figure 2) was the highest in Mandared grafted onto C57 (225 kg), followed by Mandared grafted onto C35 (202 kg), and Mandared grafted onto C54 (195 kg).

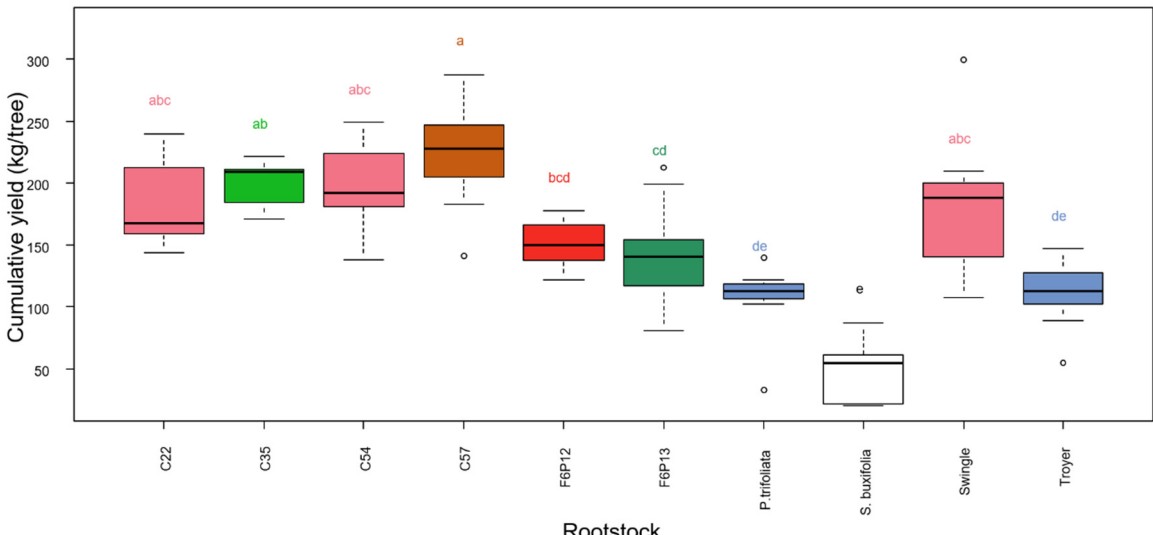

**Figure 2.** Cumulative yield of Mandared grafted onto 10 rootstocks, calculated from 2015 to 2019. Different letters indicate significantly different means according to Tukey's test at $p < 0.05$. Different colors of the boxes indicate different significance groups.

The lowest cumulative production was observed in trees on *S. buxifolia*. Interestingly, trees on C57 had double the cumulative yield of trees on Troyer, which was included in the trial as a standard rootstock. Mandared grafted onto Swingle showed high productivity (182 kg of fruit produced), but also the highest standard deviation (65 kg), indicating a high variability among replicates.

Canopy volume was significantly affected by the rootstock (Figure 3). The highest canopy volume was observed for Mandared grafted onto C57 (27.3 m³), while the lowest was observed for Mandared grafted onto *P. trifoliata* (12.8 m³), indicating the poor adaptability of trifoliate orange to the pedological conditions of the trial.

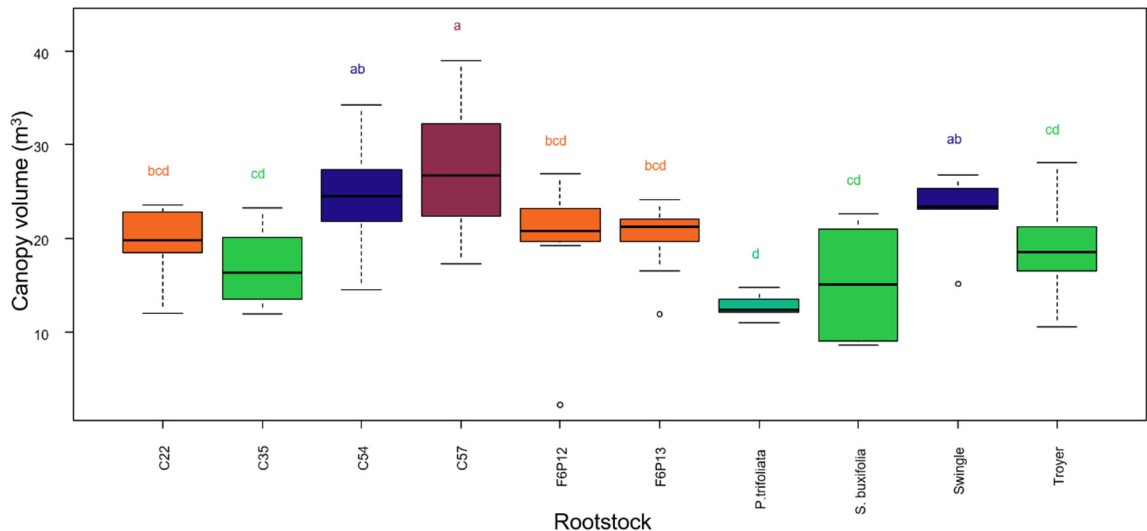

**Figure 3.** Canopy volume of Mandared grafted onto 10 rootstocks, recorded in spring 2018. Different letters indicate significantly different means according to Tukey's test at $p < 0.05$. Different colors of the boxes indicate different significance groups.

The scatterplot in Figure 4 shows the relationship between cumulative yield and canopy volume of each replicate tree. Most replicates of the best performing rootstocks fall above the regression line.

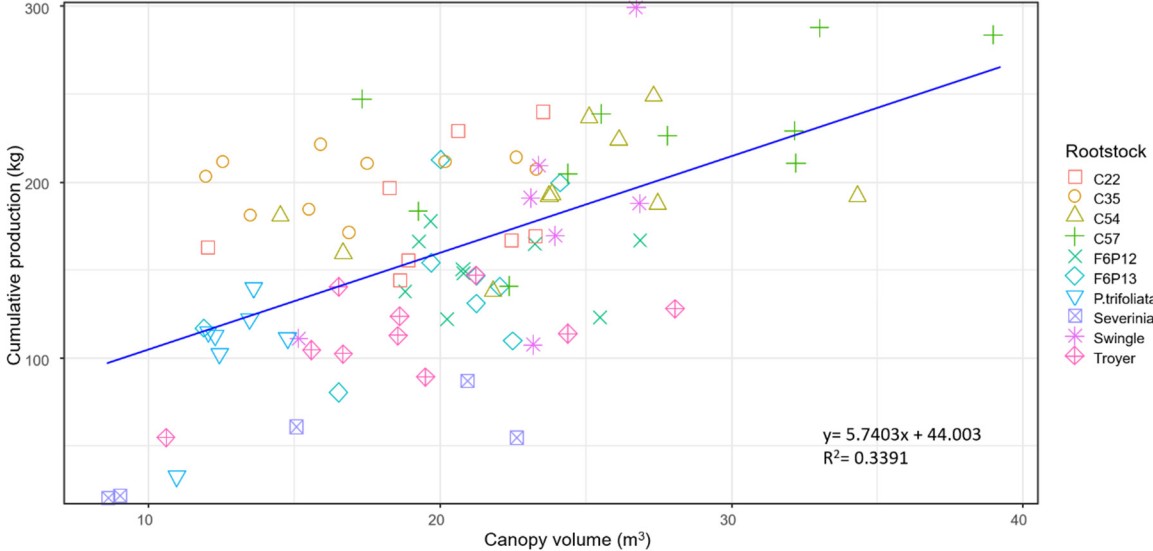

**Figure 4.** Scatterplot with regression line showing the relationships between canopy volume and cumulative production of the single replicate trees.

Mandared grafted onto C35 was the most efficient (12.4 kg m$^{-3}$; Figure 5) followed by Mandared grafted onto C22 (9.5 kg m$^{-3}$), while Mandared grafted onto *S. buxifolia* was the least efficient (3.1 kg m$^{-3}$).

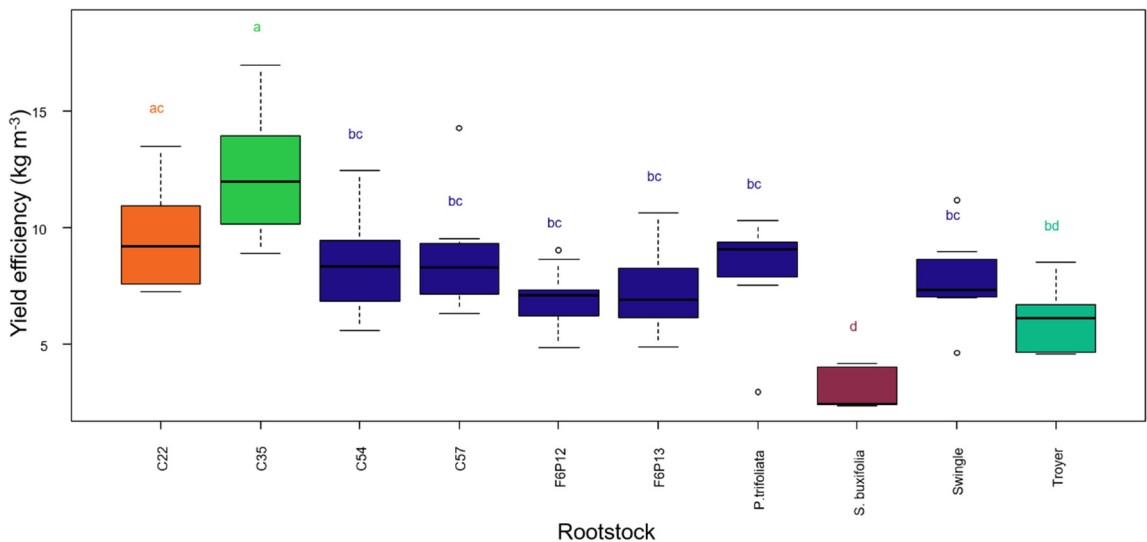

**Figure 5.** Yield efficiency of Mandared grafted onto 10 rootstocks. Different letters indicate significantly different means according to Tukey's test at $p < 0.05$. Different colors of the boxes indicate different significance groups.

The differences in fruit weights among rootstocks were not statistically significant (Figure 6). The average fruit weight was highest for trees on C35 and Swingle (183.6 and 183.9 g, respectively).

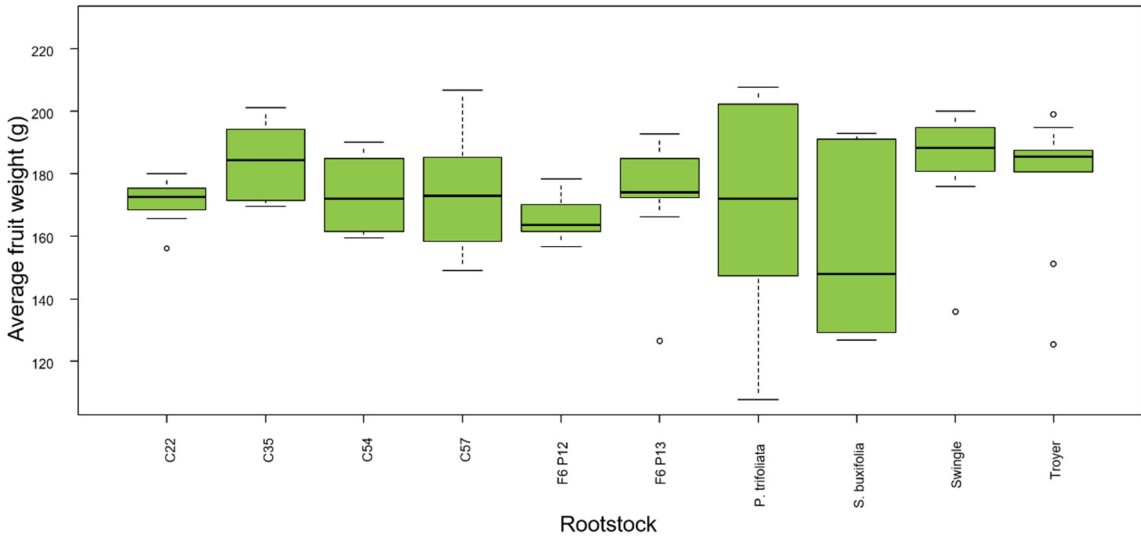

**Figure 6.** Fruit weight of Mandared grafted onto 10 rootstocks.

Despite the use of triclopyr and 2, 4-D, we observed a marked pre-harvest fruit drop. This was partly due to thorn damage and Mediterranean fruit fly attacks, but mostly attributable to the cultivar characteristics. However, the percentage of fruit drop varied among rootstocks and among years. Huge differences were found in the percentage of fruit drop of Mandared grafted onto different rootstocks (Figure 7).

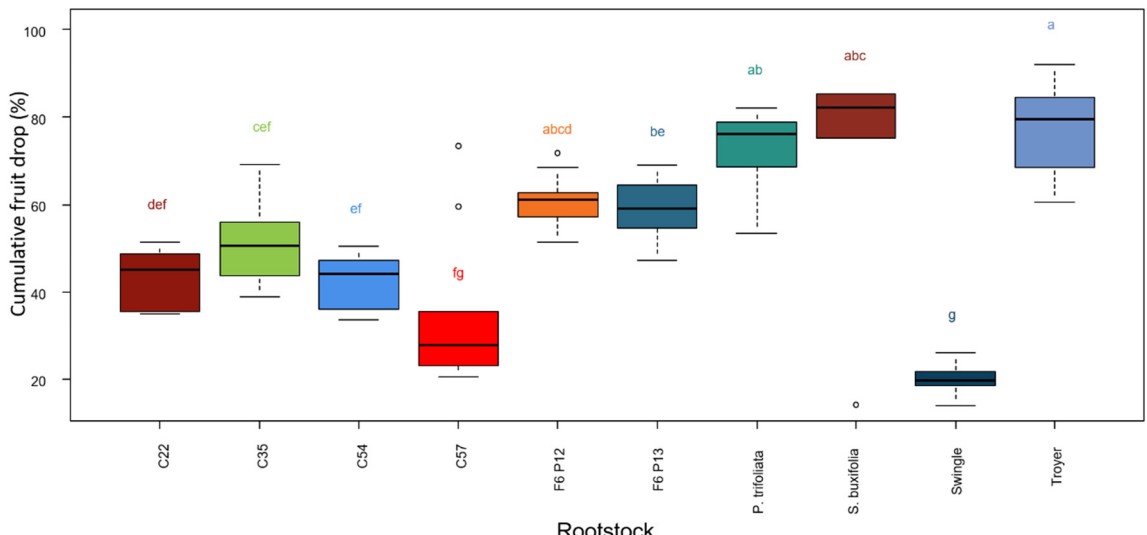

**Figure 7.** Percentage of fruit drop in Mandared grafted onto 10 rootstocks, recorded between 2016 and 2019. Different letters indicate significantly different means according to Tukey's test at *p* < 0.05. Different colors of the boxes indicate different significance groups.

The trees on Swingle showed the highest ability to retain fruits on the tree, with an average of fruit drop of 20.0%, followed by the trees on C57, with an average fruit drop of 34.6%. Intermediate fruit drop rates were observed in Mandared grafted onto C54 (42.5%), C22 (43.1%), and C35 (51.0%), while the highest rates were found in Mandared grafted onto *P. trifoliata* (72.4%) and Troyer (77.1%). Regarding the rates of fruit drop in different years, we observed particularly severe events in 2017 and 2019, associated with two frost events that occurred a few days before harvest. Specifically, in early January 2017, we recorded two consecutive nights when temperatures were below 0 °C for at least 8 h, and minimum temperatures reached −4 °C and −3 °C for approximately 1 h, respectively. In 2017, the average fruit loss was close to 100% in Mandared grafted onto *P. trifoliata* and Troyer, 82% in Mandared grafted onto F6P12 and F6P13, and 73% in Mandared grafted onto C35, while the average fruit loss in trees on Swingle was less than 30%. A similar event occurred in 2019, with the minimum temperature reaching −2.5 °C for one night. In 2019, pre-harvest fruit drop in Mandared grafted onto Swingle was approximately 32%, while it was 76% in trees on C35, and fruit loss close to 100% was observed in Mandared grafted onto Troyer, *P. trifoliata*, F6P12, and F6P13. In 2016 and 2018 there were no frost event before harvest, and rates of fruit drop were less severe (Figure S1). However, the trees on Swingle and C57 always showed the best performance with regards to reducing the fruit drop rate.

Fruit quality was analyzed in January 2017 and 2019 (Table 3).

**Table 3.** Qualitative analysis on Mandared fruits sampled in 2017 and 2019. Different letters indicate significant differences among rootstocks in each year as determined by Tukey multiple range test (*p* ≤ 0.05).

| Rootstock | TSS (°Brix) | | Acidity (g L$^{-1}$) | | TSS/Acidity | | Juice (%) | | Anthocyanins (mg L$^{-1}$) | | Peel Color Index | |
|---|---|---|---|---|---|---|---|---|---|---|---|---|
| | 2017 | 2019 | 2017 | 2019 | 2017 | 2019 | 2017 | 2019 | 2017 | 2019 | 2017 | 2019 |
| C22 | 11.97 | 12.07 | 14.93 | 13.34 | 8.03 | 9.32 | 56.67 | 57.24 ab | 10.47 a | 9.88 | 8.59 a | 6.61 |
| C35 | 12.07 | 10.53 | 15.70 | 14.62 | 7.70 | 7.24 | 56.33 | 56.49 ab | 3.16 b | 10.23 | 7.56 b | 6.43 |
| C54 | 10.87 | 11.03 | 13.97 | 13.93 | 7.80 | 7.95 | 54.67 | 59.87 a | 3.79 ab | 5.26 | 7.74 ab | 7.66 |
| C57 | 11.00 | 11.57 | 14.70 | 13.16 | 7.47 | 8.86 | 59.40 | 56.07 ab | 3.44 ab | 4.87 | 7.48 b | 6.46 |
| Swingle | 11.43 | 12.27 | 15.33 | 15.76 | 7.47 | 7.09 | 60.20 | 56.66 ab | 3.87 ab | 7.21 | 7.72 ab | 5.83 |
| F6P12 | 10.80 | 12.27 | 14.73 | 12.33 | 7.40 | 9.97 | 52.53 | 54.72 b | 6.8 ab | 10.52 | 7.89 ab | 6.52 |
| F6P13 | 11.40 | 10.73 | 14.97 | 15.27 | 7.63 | 7.19 | 55.50 | 54.92 ab | 4.96 ab | 4.19 | 8.07 ab | 8.56 |
| *P. trifoliata* | 10.80 | 10.13 | 15.47 | 12.56 | 7.03 | 8.06 | 54.13 | 54.27 b | 1.88 a | 7.77 | 7.49 b | 6.37 |
| *S. buxifolia* | 11.57 | 10.40 | 16.10 | 11.95 | 7.20 | 8.73 | 49.13 | 58.38 ab | 5.60 ab | 9.16 | 8.24 ab | 6.23 |
| Troyer | 11.90 | 11.70 | 15.23 | 13.40 | 7.83 | 8.74 | 58.43 | 59.30 ab | 5.07 ab | 5.87 | 7.92 ab | 6.18 |

Fruits were carefully selected and fruits that were damaged due to low temperatures were discarded. TSS ranged from 10.13 (Mandared grafted onto *P. trifoliata*) to 12.27 °Brix (Mandared grafted onto F6P12 and Swingle), and acidity ranged from 11.95 to 16.10 g $L^{-1}$. However, differences related to TSS and acidity were not statistically different (Tukey multiple range test; $p > 0.05$). The lack of significance might be due to an insufficient number of sampled fruits and/or replicate trees considered in the analysis. Significant differences related to anthocyanin accumulation, peel color index, and juice percentage were observed in one sampling year. Specifically, significant differences were found in juice percentage in 2019, and in anthocyanins and peel color index in 2017. In 2019, the TSS/acidity ratio was usually higher than in 2017. In 2019, the highest average TSS/acidity ratios were observed in Mandared grafted onto F6P12, C22, and C57.

## 4. Discussion

### 4.1. Effect of Rootstocks on Yield Precocity, Productivity, and Fruit Quality

In our study we evaluated the productivity of Mandared triploid mandarins in combination with 11 rootstocks during the first five years of production. The scion grafted onto different rootstocks showed significant differences in terms of yield precocity, cumulative production, and yield efficiency. Some fruit traits also varied among rootstocks, but only in one of the two sampling years in which fruit quality was analyzed.

Mandared grafted onto the rootstocks of the UCR breeding program (in particular C22, C57, and C35) started to set fruits one year earlier than Mandared grafted onto the other rootstocks. This result was previously reported by Continella et al. [3], who performed a similar trial in combination with Scirè Tarocco blood orange. The four UCR rootstocks were among the most productive. Moreover, C35 and C22 showed high yield efficiencies. The good performance of the new UCR rootstocks was previously reported in studies performed in the United States [17,18], and these rootstocks appeared promising for use in the Mediterranean area [3]. The trees on Troyer, which is among the most diffused rootstocks worldwide [2] did not perform well, with about a half of the productivity the trees on C57. This clearly indicates that low productivities sometimes encountered in new citrus varieties may be corrected by choosing the appropriate rootstock. Mandared grafted onto the CREA hybrids (F6P12 and F6P13), which in previous trials showed high productivity [8], were not among the most productive rootstocks. Moreover, the trees on F6P12 and F6P13 delayed their first production, so it is not advisable to use these rootstocks in combination with Mandared. The trees on Swingle showed high productivity, but also had the highest variation among replicates. This variability may be attributable to the relatively high levels of active lime in some areas of the experimental farm at which our study was conducted. Swingle is known to be sensitive to iron chlorosis [19] when active lime levels are around or above 4% [8,11], and iron chlorosis might have affected productivity.

Regarding fruit quality traits, statistically significant differences related to TSS and acidity were not observed. Mandared showed a level of TSS comparable to other mandarin cultivars but a higher acidity. The Mandared high acidity was already reported in a previous study [10] and confers a tart taste similar to that of Tarocco blood orange, which is the male parent of this hybrid. Some fruit traits, such as anthocyanin concentration and juice percentage also differed among rootstocks, at least in one sampling year. Anthocyanin accumulation is known to be a critical trait for high quality fruits. The rate of anthocyanin accumulation is rather complex and variable among pigmented cultivars [11]. In addition to the genotype, this trait is highly affected by the environmental (low temperature during winter) and culture conditions, as well as by the rootstock [3]. In our study, fruits of Mandared grafted onto C22 had the highest anthocyanin accumulation in 2017, and had a similar anthocyanin accumulation in 2019, while the year-to-year variability of the Mandared grafted onto the other rootstocks was higher. It would be worthwhile to investigate the rate of anthocyanin accumulation of Mandared grafted onto C22 in the following years, since a rootstock conferring high and stable anthocyanin accumulation would be desirable for the citrus industry. It will be also important to perform future investigations

on a larger number of fruit samples during several years to clarify the influence of the rootstock on pigmented mandarin quality.

### 4.2. Influence of Rootstocks on Pre-Harvest Fruit Drop

Pre-harvest fruit drop is a phenomenon that is exacerbated by biotic and abiotic stresses [20,21] and is influenced by the scion genotype. Blood oranges, but also mandarin hybrids such as Kinnow, are known to be particularly sensitive to pre-harvest fruit drop [22,23]. Fruit drop is one of the main disadvantages of Mandared. This characteristic is typical of Tarocco, which is the tetraploid pollen parent of Mandared. In Mandared, we noticed that the tendency to fruit drop is even stronger than in Tarocco, especially in the case of frost events.

An effect of the rootstock on pre-harvest fruit drop has been observed in previous studies [24–26]. In the present trial we noticed a strong effect of the rootstock on rates of fruit drop, which was particularly significant in cases of frost events occurring before harvest. In particular, Mandared grafted onto Swingle and C57 showed the best performance, and both might be suggested to reduce pre-harvest fruit drop in Mandared, whilst maintaining high productivity and fruit quality.

In conclusion, C57, C35, and C22 were the best performing rootstocks in our conditions (Figure 8).

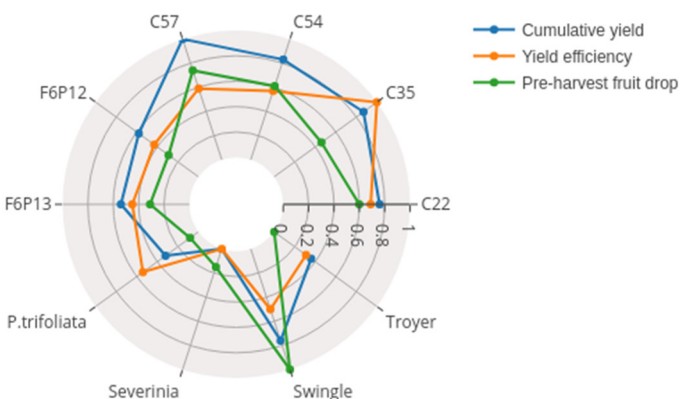

**Figure 8.** Radar chart showing the yield performance of the 10 rootstocks in combination with Mandared. Data were rescaled independently, normalizing the best performance to 1 and the worst performance to 0 for each parameter.

Mandared onto these three rootstocks started to set fruits one year earlier than the others. The trees grafted onto C35 showed some of the highest production efficiencies, and started to produce fruit one year earlier than Mandared grafted onto standard rootstocks such as Troyer and Swingle. These results have implications for the cultivar profitability. The trees on Swingle had the strongest ability to avoid pre-harvest fruit drop, but also the highest variability among replicates. Mandared grafted onto C57 showed a significant reduction in rates of fruit drop, and resulted in one of the highest cumulative production. Mandared grafted onto C22 showed yield precocity and high yield efficiency, comparable to that of Mandared grafted onto C35. The reduced canopy volume of trees on C22 has been observed in previous studies [3,27], and could be an advantage for new plantings with higher densities. In previous studies, C22 has been shown to be particularly tolerant to high pH and soils with a high percentage of active lime [17] as well as being tolerant to conditions of high salinity [28]. We tested this rootstock in a soil with moderate levels of active lime, where trees on F. dragon, *P. trifoliata*, and Swingle showed iron chlorosis symptoms. However, the conditions at our field site were not amongst the most extreme in terms of active lime percentages. Due to its overall good performance, and due to the need for alternatives to sour orange in high pH soils, it would be worthwhile to test C22 in soils with higher levels of active lime, which are often found in Italy and other Mediterranean citrus producing countries,

such as Spain and Turkey. Taken together, the above results provide essential new information for the selection of rootstocks for new citrus orchards.

**Supplementary Materials:** The following is available online at http://www.mdpi.com/2073-4395/10/9/1305/s1, Figure S1: Percentage of fruit drop recorded in 2016, 2017, 2018 and 2019. Different letters indicate significantly different means according to Tukey's test at $p < 0.05$.

**Author Contributions:** Conceptualization, investigation, formal analysis, writing—original draft, M.C.; conceptualization, investigation, funding acquisition, writing—review and editing, A.C.; investigation, G.M., C.P., R.R., F.S., C.A.; writing—review and editing, A.G.; conceptualization, investigation, funding acquisition, G.R. All authors have read and agreed to the published version of the manuscript.

**Funding:** This research received no external funding.

**Acknowledgments:** We are grateful to Mikeal Roose, Claire Federici and Robert Krueger for providing the seeds of the UCR rootstocks, to Michele Scirè for management of the rootstock trial at the Palazzelli experimental farm, to Angelo Ciacciulli for the support in the data analysis, to Filippo Ferlito for providing the climatic data, and to Giuseppe Reforgiato Recupero for the help in establishing the trial and sampling the fruits.

**Conflicts of Interest:** The authors declare no conflict of interest.

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
