# Peer review of "Rootstocks Influence Yield Precocity, Productivity, and Pre-Harvest Fruit Drop of Mandared Pigmented Mandarin"

_agronomy, doi:10.3390/agronomy10091305_

Round 1

Reviewer 1 Report

The paper is interesting, well written and presented, with high importance to growers. However, a revision is required in regard to statements made with incompliance with statistical analysis results.

L 60 Add ref after “pigmentation”.

L67 Explain CREA initials

L208 Please rephrase, if differences were not statistically significant, a statement like “UCR rootstocks generally had fruits with lower weights” can not be made. It can only be made if the authors cluster all UCR and compare to all others and the differences are significant.

Please revise Y axis for all figures, e.g. Fig 5 change Y axis to “Fruit size (g)”; Fig 6 change Y axis to “Fruit drop (%)”.

L222 The sentence starting with “Huge differences were found” does not seem to be in context, rephrase or omit.

Table 2 Please add units to TSS, change colour to color to be consistent with text. Significantly higher values should be marked with “a”, while lower values should be marked with “b” (e.g. in Peel color index).

L261 Please rephrase sentence starting with “however, differences among means”

L262 Sentence starting with “significant differences related to…” is unclear, please revise.

L279 “The four UCR rootstocks also resulted in the highest yields”- both statements in this sentence are not supported by a proper statistical analysis, either rephrase, or redo the statistical analysis.

L285 Sentence starting with “‘Mandared’ grafted onto the CREA hybrids”- again, this statement is not supported by statistically significant differences.

L297 “Variability was also found”- variability can not ne statistically insignificant.

Section 4.1 please compare TSS acidity and their ratio to general accepted values in mandarins.

L310 Rootstock or scion genotype?

L321 C35 is not significantly different from C22.

L324 “but fruits were generally less pigmented and slightly more acidic”- this is not statistically significant.

L326 “and resulted in the highest cumulative production” C57 is not significantly different from C22, C35 and C54.

L328 “and higher than ‘Mandared’ grafted onto C57”- not statistically significant.

L331 Sentence starting with “Moreover, fruits of ‘Mandared”- C22 anthocyanin concentration and peel CI are not significantly highest.

L 285-L 290 and L321-L332 should be revised to comply with statistical analysis.

A suggestion is made to consider summarizing all (or main) findings by presenting them in a spider web figure, which will show for each rootstock its pros and cons, and will thus allow an easier comparison between the examined rootstocks.

Author Response

Response to reviewer 1

The paper is interesting, well written and presented, with high importance to growers. However, a revision is required in regard to statements made with incompliance with statistical analysis results.

We thank reviewer 1 for considering our paper well written and presented, and for giving us useful suggestions about statistical significance of some of the presented data. His opinion helped to improve the new version of the manuscript.

The detailed changes based on the reviewer’s suggestions are listed below. Due to the modifications in the manuscript, the line numbering has changed.

L 60 Add ref after “pigmentation”.

We added a reference as requested

L67 Explain CREA initials

We added CREA full name as requested

L208 Please rephrase, if differences were not statistically significant, a statement like “UCR rootstocks generally had fruits with lower weights” can not be made. It can only be made if the authors cluster all UCR and compare to all others and the differences are significant.

We deleted the sentence since the statement was not supported by statistical significance

Please revise Y axis for all figures, e.g. Fig 5 change Y axis to “Fruit size (g)”; Fig 6 change Y axis to “Fruit drop (%)”.

We revised the Y axis labels of all figures

L222 The sentence starting with “Huge differences were found” does not seem to be in context, rephrase or omit.

We changed the sentence, since there was a mistake in the first version of the manuscript. The correct sentence is “Huge differences were found in the percentage of fruit drop of ‘Mandared’ grafted onto different rootstocks (Figure 7).”

Table 2 Please add units to TSS, change colour to color to be consistent with text. Significantly higher values should be marked with “a”, while lower values should be marked with “b” (e.g. in Peel color index).

We added units to TSS, and changed “colour” to “color”. We changed the letters of significance as requested.

L261 Please rephrase sentence starting with “however, differences among means”

We changed the phrase to “However, differences related to TSS and acidity were not statistically different”.

L262 Sentence starting with “significant differences related to…” is unclear, please revise.

We changed the sentence to “Significant differences related to anthocyanin accumulation, peel color index, and juice percentage were observed in one sampling year. Specifically, significant differences were found in juice percentage in 2019, and in anthocyanins and peel color index in 2017”.

L279 “The four UCR rootstocks also resulted in the highest yields”- both statements in this sentence are not supported by a proper statistical analysis, either rephrase, or redo the statistical analysis.

We rephrased the sentence as suggested. The new version is “The four UCR rootstocks were among the most productive. Moreover, C35 and C22 showed high yield efficiencies. The good performance of the new UCR rootstocks was previously reported in studies performed in the United States”.

L285 Sentence starting with “‘Mandared’ grafted onto the CREA hybrids”- again, this statement is not supported by statistically significant differences.

We rephrased the sentence into “ ‘Mandared’ grafted onto the CREA hybrids (F6P12 and F6P13), which in previous trials showed high productivity [7], were not among the most productive rootstocks.”

L297 “Variability was also found”- variability can not be statistically insignificant.

We deleted the sentence

Section 4.1 please compare TSS acidity and their ratio to general accepted values in mandarins.

We added a sentence regarding the values of TSS and acidity: “Regarding fruit quality traits, statistically significant differences related to TSS and acidity were not observed. ‘Mandared’ showed a level of TSS comparable to other mandarin cultivars but a higher acidity. The ‘Mandared’ high acidity was already reported in a previous study [9] and confers a tart taste similar to that of ‘Tarocco’ blood orange.”

L310 Rootstock or scion genotype?

We added “scion”

L321 C35 is not significantly different from C22.

We changed the sentence to “The trees grafted onto C35 showed some of the highest production efficiencies”

L324 “but fruits were generally less pigmented and slightly more acidic”- this is not statistically significant.

We deleted the sentence, since it was not supported by statistical significance

L326 “and resulted in the highest cumulative production” C57 is not significantly different from C22, C35 and C54.

We changed the sentence to “Furthermore, ‘Mandared’ grafted onto C57 showed a significant reduction in rates of fruit drop, and resulted in one of the highest cumulative production.”

L328 “and higher than ‘Mandared’ grafted onto C57”- not statistically significant.

We deleted “and higher than ‘Mandared’ grafted onto C57”.

L331 Sentence starting with “Moreover, fruits of ‘Mandared”- C22 anthocyanin concentration and peel CI are not significantly highest.

We deleted the sentence because not supported by statistical significance

L 285-L 290 and L321-L332 should be revised to comply with statistical analysis.

We rephrased this part deleting parts which were not supported by statistical analysis

A suggestion is made to consider summarizing all (or main) findings by presenting them in a spider web figure, which will show for each rootstock its pros and cons, and will thus allow an easier comparison between the examined rootstocks.

We generated a radar chart (Figure 8) to summarize the yield performances of the analysed rootstocks

Reviewer 2 Report

This is a well designed and executed rootstock scion rootstock scion study that produces results for these rootstocks consistent with reports with other scions in the literature.   Very complete rootstock/scion valuation of a potential mandarin cultivar.  

Bot both the plural and singular form is are "damage"; damage os not made plural by adding an s.

What is new is the evaluation of three new rootstocks, C-22,54 and 57.  The first, C-22 appears t have promise for higher pH soils and effects on tree size, yield efficiency and productivity..  The major weakness of this MS is that the scion appears has poor potential as a commercial mandarin of this scion; information which became more evident as the trial progressed.

Author Response

We are grateful to reviewer 2 for his positive evaluation of our paper. We agree that the major weakness of the study is the behaviour of scion. However, the scion weakness helped to evaluate the positive effects of some rootstocks.

We eliminated the plural form of “damage” throughout the manuscript.

Reviewer 3 Report

The manuscript deals with the comparison of several citrus rootstocks for cultivation of pigmented mandarins in Mediterranean semi-arid areas, and specifically in calcareous soils. The manuscript is generally well written and organized and deserves to be published. There are just a couple of major concerns, one regarding the statistical data analysis, which doesn't seem to reflect the experimental design; the other regarding the fruit quality measurements, which were done in only 2 out 5 years, the two years showing very low crop loads due to frost damages. As indicated in the attached file, the Authors should reconsider data analysis using the year as a (fixed or random) factor in the ANOVA model; cumulative data may also be shown and analyzed by Analysis of Means. Quality data in the present form are not representative of those trees and conditions, so they should be presented and discussed with extreme caution as no conclusive statement can be done at this time, but future investigation should reveal the real effect of the rootstock on pigmented mandarin quality. These and other minor comments are marked in the attached pdf file, and must all be properly accommodated before publication.

Author Response

Response to reviewer 3

We are grateful to reviewer 3 for the useful suggestions for the manuscript improvement. We replied to all his questions and concerns. We modified major and minor points in the manuscript, that are visible with the track changes. We better clarified the experimental design and the ANOVA model. We also added a table regarding the yield for the single years and a scatterplot indicating the relationship between canopy volume and cumulative production, as requested.

Regarding the effect of frost of fruit quality, we added a sentence in which we clearly indicated that further research is needed to understand the effect of rootstocks on fruit quality of pigmented mandarins.

Below you will find the list of modification, and replies to concerns and doubts of the reviewer.

Line 3: we replaced “triploid” with “pigmented”

Abstract: we shortened the abstract as suggested

Line 86: we specified the full name of CREA

Line 90: Clementine is correct with capital letter

Line 110-114: We included the personal communication from one of the breeders who released the cultivar, who is also one of the authors of the present manuscript. Unfortunately, as for other citrus cultivars, many positive and negative characteristics of ‘Mandared’ were observed after its release, and there are no published reports regarding these aspects.

Line 121: we changed the sentence as suggested, making it more specific

Line 126: we added “Italy” to the location

Line 127: we replaced “a spacing of” with “trees spaced at”

Line 134: Correct, we are dealing with an unreplicated block. We changed the sentence into: “The trial was established in July 2010 in a completely randomized design using 10 single tree replicates for each rootstock”

Line 136: we made a mistake in the first version, writing “experimental farm”. We replaced it with “experimental block”. We prefer to leave the range of active lime and not the average, because we observed high variability in the active lime concentration of the block, due to stripes of calcareous soil. The range gives a better view of the non-homogeneous soil conditions.

Line 137: as suggested, we replaced “subjected to standard agronomical practices” with “grown under conventional cultural practices”

Line 139: “fruits were treated once”

Line 140: we deleted “another”

Line 163: fruit drop counts were performed few days before harvest, so only mature fruits were considered. We added the sentence “(10 days and one day before, respectively)” to specify this aspect.

Line 172: We included “15 fruits per plant”, that we did not specify in the previous version

Line 195: We specified the ANOVA model as suggested. The sentence was changed into “Comparisons of means were performed using one-way analysis of variance (ANOVA) with Tukey tests and the confidence level was at 95%, with rootstock genotypes as fixed effects”. Rootstock effect (fixed factor) was evaluated on different parameters. The analyzed data were: cumulative yield (no year effect, but sum of 2015-19 productions); production of each year as specifically requested by the reviewer (see the next review point); canopy volume (recorded in 2018, year of the maximum canopy growth); yield efficiency (recorded in 2018, year of the maximum yield and canopy growth); and fruit qualitative characteristics analyzed per year (2017 and 2019).

Line 224 and line 228: we included a table (table 2) indicating the cumulative production and statistics for each year. In this way it is possible to evidence the shorter unproductive period of some rootstocks

Figures (box plots) and data visualization: box plots can be applied to data with normal and non-normal distributions. This method was chosen because in our opinion it better highlights the variation and the distributions of replicates of each rootstock, and evidences the presence of outliers. At the same time, the results of the Tukey test were indicated, so the data visualization appears complete. Different colours of the boxes indicate different significance groups, to facilitate readers. We included this information in the captions. We changed the significance letters in all figure, giving “a” to the highest means. We also added the Y-axis titles that were missing, and deleted the letters where statistical significance was absent

Line 268: as suggested by the reviewer, we generated a scatterplot (figure 4) showing the relationship between canopy volume and production including the regression line, so the best performing rootstocks are evident.

Line 275: “was the most efficient”

Lines 276-277: “Mandared grafted onto S. buxifolia was the least efficient”

Line 285: As suggested, we modified this paragraph inverting the order of the sentences. Since there was no statistical significance among means, we decided to delete the part related to the UCR rootstocks.

Line 308: “in the percentage of fruit drop”

Line 343: we are aware that the two-year analysis performed in 2017 and 2019 gives a partial view of the fruit quality. However, as already indicated, all damaged fruits were discarded before performing the analysis. Also, fruit drop occurred few days before harvest so we believe that the effect of the heavy fruit drop on the remaining fruits is negligible, although this would require a specific research. Regarding this concern of the reviewer, we added a sentence at lines 416-418: “. It will be also important to perform future investigations on a larger number of fruit samples to clarify the influence of the rootstock on pigmented mandarin quality.”

Table 3: Letter “a” was assigned to the highest means as suggested.

Line 354-356: the lack of significance for different qualitative traits might be due to a low number of samples and/or replicate trees considered. We added a sentence about that: “The lack of significance might be due to an insufficient number of sampled fruits and/or replicate trees considered in the analysis”. We don’t believe that the lack of significance is related to fruit drop, since many rootstock trials described in bibliography showed no differences among rootstocks in terms of fruit qualitative features.

Line 407: differed

Line 408: we deleted the sentence based on the suggestion of reviewer 1

Line 410-411: Unfortunately, there are no scientific reports on the effects of daily temperature fluctuations on fruit pigmentation. The only proved effects are related to cold exposure.

Round 2

Reviewer 1 Report

The authors did a very good job on improving the MS, and it is ready for publication in Agronomy. 

For next, please add the new line numbers where changes have been made. Without line numbers it's difficult for the reviewer to follow.